# Clinical and pathological features of thrombotic microangiopathy influencing long-term kidney transplant outcomes

**Cínthia Montenegro Teixeira**[1]*, **Hélio Tedesco Silva Junior**[1], **Luiz Antônio Ribeiro de Moura**[2], **Henrique Machado de Sousa Proença**[2], **Renato de Marco**[3], **Maria Gerbase de Lima**[3], **Marina Pontello Cristelli**[1], **Laila Almeida Viana**[1], **Cláudia Rosso Felipe**[1], **José Osmar Medina Pestana**[1]

**1** Nephrology Division, Federal University of São Paulo, São Paulo, Brazil, **2** Pathology Division, Hospital do Rim, Federal University of São Paulo, São Paulo, Brazil, **3** Immunogenetics Institute, AFIP, São Paulo, Brazil

* cinthiamt@gmail.com

**Data Availability Statement:** All relevant data are within the manuscript.

## Abstract

### Introduction

Thrombotic microangiopathy (TMA) in post-transplant setting has heterogeneous clinical manifestations.

### Methods

We retrospectively studied data of 89 patients with post-transplant TMA, which was characterized by thrombi in at least one glomerulus and/or arteriole. Systemic TMA was defined by thrombocytopenia and microangiopathic anemia and early onset TMA, when occurred less than 90 days post transplant.

### Results

The cumulative incidence was 0.93%. The majority of the recipients were young (mean age 39 years), female (52%) and Caucasian (48%) with primary kidney disease of unknown etiology (37%). Early TMA occurred in 51% of the patients and systemic TMA, in 25%. Underlying precipitating factors were: infection (54%), acute rejection (34%), calcineurin inhibitor toxicity (13%) and pregnancy (3%). 18% of the patients had several triggers. Glomerular TMA was observed in 50% of the biopsies and endothelial cell activation, in 61%. The 1-year patient survival was 97% and corresponding graft survival, 66%. Allograft survival was inferior when acute antibody mediated rejection (ABMR) occurred (with 41%; without 70%, p = 0.01), however no differences were determined by hemolysis, time of onset, thrombi location or endothelial cell activation.

### Conclusions

Our results suggest that post-transplant TMA is a rare but severe condition, regardless of its clinical and histological presentation, mainly when associated to ABMR.

**Funding:** This study was supported by CAPES - "Coordenação de Aperfeiçoamento de Pessoal de Nível Superior". The funders had no role in study design, data collection and analysis, decision to publish, or preparation of the manuscript.

**Competing interests:** I have read the journal's policy and the authors of this manuscript have the following competing interests: Dr. HTS reports grants and personal fees from NOVARTIS, grants and personal fees from PFIZER, grants and personal fees from SANOFI and grants from QUARK, outside the submitted work. This does not alter our adherence to PLOS ONE policies on sharing data and materials. All the other authors declared no relevant competing interests.

**Abbreviations:** TMA, thrombotic microangiopathy; ABMR, antibody mediated rejection; aHUS, atypical hemolytic uremic syndrome; CNI, calcineurin inhibitor; IF/TA, interstitial fibrosis and tubular atrophy; TCMR, acute cellular rejection; TTP, thrombocytopenic thrombotic purpura; eGFR, estimated glomerular filtration rate; CKD-EPI, chronic kidney disease epidemiology collaboration; DIC, disseminated intravascular coagulation; PRA, panel-reactive antibodies; mTOR, mammalian target of rapamycin; AT1R, angiotensin II type 1 receptor; HELLP, hemolysis, elevated liver enzymes and low platelets; HLA, human leukocyte antigen; PNH, paroxysmal nocturnal hemoglobinuria (PNH).

## Introduction

Thrombotic microangiopathy (TMA) is defined histologically by the presence of arteriolar and/or glomerular thrombosis [1] and is a hallmark of a broad spectrum of diseases that affect the vascular endothelium. After kidney transplantation, the incidence of TMA varies between 0.8% and 14% [2–6] and occurs as a recurrence of Atypical Hemolytic Uremic Syndrome (aHUS) or as *de novo* disease. Although its histological features are well defined, the clinical etiological diagnosis is challenging because TMA may be associated with several triggers, involving genetic susceptibility [7–9] and environmental factors, such as ischemia, antibody-mediated rejection, calcineurin inhibitor (CNI) toxicity and infection [10]. TMA clinical manifestation is also heterogeneous, varying from life-threatening systemic hemolysis to lesions restricted to the dysfunctional allograft. Therapeutic options include temporary or definitive CNI withdrawal, plasma exchange therapy, treatment of the underlying triggering factor(s), and use of complement system blockers [11]. Overall, post-transplant TMA has been associated with poor allograft outcomes with up to 40% of graft loss [2,12].

Considering the TMA significant negative impact on graft survival, advances in the understanding of its clinical presentation, underlying pathogenesis and prognostic features is fundamental to devise more effective and safety preventive and therapeutic strategies. Previous studies in children with HUS revealed that specific histological lesions in native kidney predicted development of chronic kidney disease [13–18]. In the post-transplant setting, it remains unclear whether the TMA histological patterns and clinical presentation have distinct pathogenic mechanisms and, ultimately, result in different clinical outcomes. [2,3,19–20]

The aim of the present study was to present the clinical features and pathologic changes of TMA in a cohort of kidney or kidney-pancreas transplanted recipients who developed TMA, and correlate them with allograft outcomes.

## Patients and methods

### Study design and population

In this retrospective cohort study, we initially retrieved all consecutive unselected reports of renal transplant biopsies from Hospital do Rim database between January 2011 and December 2015. These biopsies were performed for graft dysfunction, new onset proteinuria or delayed graft function from kidney and kidney-pancreas transplanted patients. Of a total of 6886, we selected 119 biopsies whose reports described features of TMA. Final diagnosis was confirmed by one of the pathologist authors (LARM).

All data were fully anonymized before accessed. The protocol adheres to the 2000 Declaration of Helsinki as well as the Declaration of Istanbul 2008. The institutional review board (Comitê de Ética em Pesquisa-CEP-UNIFESP) waved the requirement for informed consent and approved this study (protocol number 1643995).

### Histological features of TMA

TMA was defined as the presence of occlusive fibrin-platelet thrombi in at least one glomerulus and/or renal arteriole/artery on renal transplant biopsies. Tissues for light microscopy were fixed in 4% formaldehyde, embedded in paraffin using routine procedure. Three to five-micrometer thick sections were cut from the tissue blocks and stained with hematoxylin-eosin, Masson's Trichrome with aniline blue, and Jones' silver staining. Acute cellular rejection and interstitial fibrosis and tubular atrophy (IF/TA) index were graded according to the Banff 13 criteria [21]. The extent of involvement of peritubular capillaries by linear deposition of C4d using the monoclonal antibody or by immunochemistry using polyclonal antibody was also

recorded and correlated with histology and donor-specific antibody for the diagnosis of ABMR.

Because morphological features, such as extent of histopathological involvement and presence of mesangiolysis, were associated with native kidney disease severity in patients with HUS [13–17], we hypothesized that TMA histological patterns may have prognostic value. Therefore, TMA lesions were classified into the following categories according to thrombi location: (1) glomerular TMA showing thrombi only in afferent or efferent arteriole or glomerular capillary; (2) arteriolar TMA showing thrombi only in arterioles or interlobular arteries; (3) glomerular/arteriolar TMA, when both glomerulus and arterioles were affected. The probable pattern of injury was also classified as (1) thrombotic lesions, when the only TMA feature was the presence of thrombi and (2) endothelial cell activation, defined by one or more of the following features: mesangiolysis, capillary necrosis, glomerular endothelial detachment, capillary wall thickening (obliterative arteriolopathy) defined as luminal occlusion with mural myxoid or fibrinoid change and thickening of the vessel wall. All biopsies were reviewed by the same pathologist for this study.

## Clinical presentation of TMA

TMA precipitating factors were retrospectively adjudicated and classified according to the following not mutually exclusive categories: (1) acute rejection: biopsy-proven acute cellular rejection (TCMR) or acute antibody-mediated rejection (ABMR) within one week; (2) infection: infectious complication within one week; (3) pregnancy; (4) CNI toxicity: improvements in allograft function when CNI withdrawal was the only intervention. When HUS or thrombocytopenic thrombotic purpura (TTP) was the cause of the primary kidney disease, TMA was considered recurrent.

Systemic or localized TMA was defined based on the presence or not of: thrombocytopenia (platelets $<150 \times 10^3$/mL) with microangiopathic hemolysis (either schistocytes on peripheral-blood smear, haptoglobin $<15$ mg/dL or lactate dehydrogenase $>1,000$ U/L) [3], at the time of the allograft biopsy diagnostic of TMA.

Finally, the timing of TMA presentation was classified as early ($\leq$90 days) or late ($>$90 days), considering that the highest risk of both *de novo* and recurrent TMA is between 3 and 6 months after transplantation [22].

## Clinical and laboratory data

Demographic baseline information, TMA presentation and management after the diagnosis were obtained by retrospective chart review. Estimated glomerular filtration rate (eGFR) was calculated by CKD-EPI (Chronic Kidney Disease Epidemiology Collaboration) equation [23].

## Outcomes variables

The outcome variables were analyzed after 12 months of follow-up after TMA diagnosis and included patient and graft survivals and renal graft function (eGFR). Allograft function was also compared at baseline (lowest serum creatinine within 3 months before TMA), at TMA diagnosis and at one-year after TMA. Patients with graft failure were considered to have an eGFR of 5 ml/min/1,73m$^2$. Causes of graft loss were collected and classified as acute rejection, IF/TA, recurrent or *de novo* glomerular disease, or thrombotic microangiopathy [24].

Comparisons of allograft function and survival were made according to the following characteristics: presence of hemolysis (Systemic vs. Localized TMA), time of onset (Early vs. Late onset TMA), association with acute ABMR, alone or combined with cellular rejection (TMA

with vs. without ABMR), thrombi location (Glomerular, Arteriolar or Glomerular/Arteriolar TMA) and pattern of renal injury (Thrombotic vs. Endothelial cell activation).

### Statistical analyses

Incidence density was estimated by dividing the number of patients that fulfilled criteria for TMA by the sum of the follow-up times for each individual at risk during the study period and reported as n/1,000 person-years. Kaplan-Meier patient survival and death-censored survival plots were used and log-rank test was performed for comparison between groups. Differences in allograft function were analyzed by two-way ANOVA (Bonferroni post-hoc test) for parametric data and Mann-Whitney or Kruskal-Wallis (Dunn-Bonferroni post-hoc test) for nonparametric data. A two-sided P-value <0.05 was considered statistically significant. All analyses were conducted with SPSS version 22 (IBM, Armonk, NY, USA) and STATA version 12.0.

## Results

### Incidence

Of 9,541 patients at risk, 119 patients were diagnosed with TMA in renal allograft during the study period. After pathology review, 10 patients did not fulfill the histopathologic TMA criteria and 4 were unavailable for review. Individuals with TMA associated to acute glomerulonephritis (n = 1), renal/arterial thrombosis (n = 4), donor kidney with thrombi due to disseminated intravascular coagulation (DIC) (n = 4), diagnosed only after allograft nephrectomy (n = 5) or whose clinical data were unavailable (n = 2) were excluded (Fig 1). Therefore, a total of 89 patients fulfilled the study criteria, yielding a cumulative incidence of 0.93% and an incidence density of 1.8 cases/1,000 person-years.

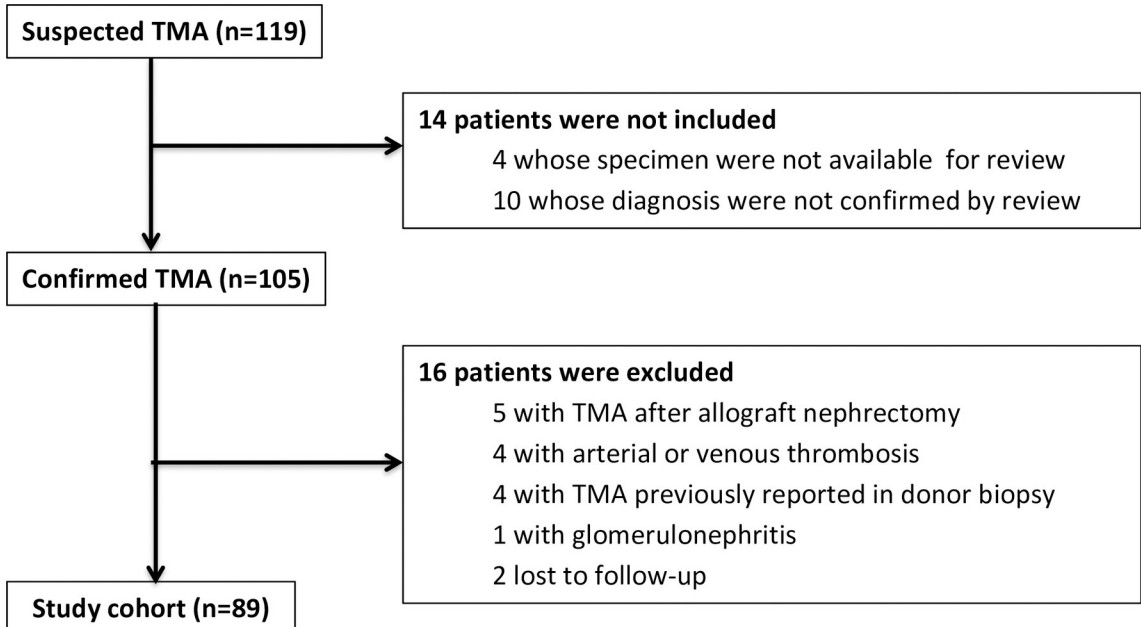

**Fig 1. Casuistic selection flowchart.** Allograft biopsies of kidney or pancreas-kidney transplanted patients with suspected TMA were reviewed by the same pathologist. Confirmed TMA was defined as the presence of occlusive fibrin-platelet thrombi in at least one glomerulus and/or arteriole. TMA = Thrombotic microangiopathy.

## Baseline characteristics

The majority of the recipients were young adults, female and Caucasian with primary kidney disease due to unknown etiology (Table 1). A high proportion of patients had pretransplant panel-reactive antibodies (PRA) Class I and Class II <50%, 90% and 92% respectively. Most individuals received the kidney transplant from a deceased-donor (68%). While 61% received induction therapy, 99% were maintained with CNI in combination with antimetabolite (87%) or mammalian target of rapamycin (mTOR) inhibitor (12%).

## TMA clinical presentation

The median time to TMA diagnosis was 3 months post transplant and 51% occurred before 3 months (Table 2). The proportion of patients receiving CNI in combination with antimetabolite decreased to 67% but did not change significantly with mTOR inhibitor (14%). At the time of TMA diagnosis, the median eGFR was 17 ml/min/1.73m$^2$ and 36% of the patients (n = 32) were on dialysis.

The most common identified triggers were infection, in 54% of the patients, and acute rejection, in 34%. CNI toxicity occurred in 13% of the patients and pregnancy, in 3%. 18% of the patients had more than one precipitating condition and, in 17%, no factor was identified. In 2% of the patients, TMA was recurrent.

Of all ABMR episodes (n = 12), 2 were associated with antibodies to angiotensin II type 1 receptor (AT1R) and one with an ABO-incompatible transplant. Urinary tract infection was the most common infection (17%) followed by cytomegalovirus and blood stream infections. The three patients who were pregnant had a TMA diagnosis with a mean gestation time of 15 weeks. Of them, two developed preeclampsia with HELLP syndrome (hemolysis, elevated liver enzymes and low platelets). The mean gestational time at delivery was 22 weeks, resulting in two abortions and one stillbirth, and two graft losses within 12 months of follow-up.

Systemic TMA occurred in 25% of the patients, being their mean hemoglobin and platelet levels 8.7±1.5g/dL and 95±46 x 10$^3$/μL, respectively, and the presence of schistocytes (95%) and reduced haptoglobin (53%) were the most common hemolysis criteria found (Table 2).

## TMA histological presentation

Glomerular TMA was the most prevalent lesion (71%), either alone (50%) or combined with arteriolar lesions (21%). Features of endothelial cell activation were observed in 61% of the biopsies. Concomitant acute cellular rejection was present in 19%. Moderate to severe IF/TA was present in 47% of the biopsy specimens (Table 3).

## Treatment after TMA

Management after TMA diagnosis was based on multiples treatments. In general, CNI withdrawal was performed in 54% of patients and plasmatherapy in 22%. In 11% of the patients, expectant management was preferred. Allograft nephrectomy was carried out in 12% of them.

Among the patients with TMA associated to rejection (N = 30), 93% received treatment for acute rejection according to the institution protocols, 37% also had the CNI withdrawal, in 20% plasmatherapy was performed and 27% also needed allograft nephrectomy due to persistent hemolysis. The patients with TMA and concomitant infection treatment (N = 48) also had supportive care as CNI cessation (60%), plasmatherapy (29%) and allograft nephrectomy (17%). Among the three pregnant patients with TMA, two had CNI withdrawal and one was subjected to plasmatherapy. The two patients with recurrent TMA were treated with plasmapheresis and CNI withdrawal, and one of them had allograft nephrectomy.

**Table 1. Baseline characteristic of TMA cases (n = 89).**

| Variables | Values |
|---|---|
| Recipient Gender, *female*, n (%) | 46 (52%) |
| Recipient Age, *years*—mean ± SD | 39 ± 14 |
| Recipient Race, n (%) | |
| *Caucasian* | 43 (48%) |
| *Mixed* | 28 (32%) |
| *Black* | 14 (16%) |
| *Others* | 4 (4%) |
| Causes of chronic kidney disease, n (%) | |
| *Glomerulonephritis* | 17 (19%) |
| *HUS or TTP* | 2 (2%) |
| *Undetermined* | 33 (37%) |
| *Diabetes Mellitus* | 13 (15%) |
| *Others* | 24 (27%) |
| [&]Time on dialysis, *months*—median (range) | 24 (0–217) |
| Historic peak PRA, n (%) | |
| Class I | |
| *0%* | 64 (71%) |
| *1–10%* | 5 (6%) |
| *11–30%* | 5 (6%) |
| *31–50%* | 6 (7%) |
| *>50%* | 9 (10%) |
| Class II | |
| *0%* | 72 (81%) |
| *1–10%* | 3 (3%) |
| *11–30%* | 5 (6%) |
| *31–50%* | 2 (2%) |
| *>50%* | 7 (8%) |
| Preexisting HLA-DSA MFI > 300, n(%) | 10 (11%) |
| HLA mismatches, (mean ± SD) | 2.5 ± 1.4 |
| Re-transplant, n (%) | 6 (7%) |
| [*]Pre-transplant AT1R antibodies (> 17UI/mL), n (%) | 16 (18%) |
| Donor age, *years*- median (range) | 47 (5–70) |
| Donor Gender, *female*, n (%) | 53 (60%) |
| Donor Race, n (%) | |
| *Caucasian* | 50 (56%) |
| *Mixed* | 28 (32%) |
| *Black* | 11 (12%) |
| Donor type, n (%) | |
| *Living* | 28 (32%) |
| *Deceased standard-criteria* | 41 (46%) |
| *Deceased expanded-criteria* | 17 (19%) |
| *Pancreas-Kidney* | 3 (3%) |
| Induction immunosuppressive treatment, n (%) | |
| *Basiliximab* | 17 (19%) |
| *Anti-thymocyte globulin* | 37 (42%) |
| *None* | 35 (39%) |

*(Continued)*

**Table 1.** (Continued)

| Variables | Values |
|---|---|
| Maintenance immunosuppressive treatment, n (%) | |
| Tacrolimus/Cyclosporine + Prednisone + AZA/MPS | 77 (87%) |
| Tacrolimus/Cyclosporine + Prednisone + Everolimus/Sirolimus | 11 (12%) |
| Others | 1 (1%) |

& out of 88 patients

* out of 87 patients

TMA = thrombotic microangiopathy; SD = standard deviation; HUS = hemolytic uremic syndrome;

TTP = thrombocytopenic thrombotic purpura; PRA = panel-reactive antibodies; HLA = human leukocyte antigens;

DSA = donor-specific antibody; MFI = mean fluorescence intensity; AT1R = angiotension II type 1 receptor;

AZA = azathioprine; MPS = mycophenolate sodium

## Outcomes

Three patients died from infectious complications, at a mean time between TMA diagnosis and death of 162 days, yielding a 1-year 97% patient survival. Corresponding graft survival was 66% (Fig 2). The primary causes of graft loss were TMA (43%), followed by acute rejection (30%) and IF/TA (24%). There was one case of focal segmental glomerulosclerosis recurrence (3%). There was allograft loss within the first 3 months post transplant in 13 patients (15%).

Allograft survival was inferior in the presence of ABMR (with 70% vs. without 41%, p = 0.01) (Fig 3). There were no statistical differences in allograft survival comparing patients with or without hemolysis (59% vs. 69%, p = 0.42), with early or late presentation (62% vs. 71%, p = 0.35), with glomerular, arteriolar or glomerular and arteriolar thrombi location (68% vs. 73% vs. 53%, p = 0.19) and with endothelial cell activation or only thrombotic lesions (63% vs. 71%, p = 0.46).

Mean eGFR was 36.9±25.9ml/min/1.73 m$^2$ at baseline, 20.6±15.5 ml/min/1.73 m$^2$ at TMA diagnosis and 28.6±23.7 ml/min/1.73 m$^2$ one year after (p<0.001, Table 4).

Hemolysis, association to ABMR, thrombi location or presence of endothelial cell activation did not correlate to behavior of allograft function in time. (Figs 4–9). Patients with a late onset TMA had a different behavior than the ones with an early TMA, given that, despite of having a higher eGFR at baseline, their allograft function did not returned to baseline after one-year of TMA diagnosis (p = 0,01). Table 4. The lower eGFR in early onset group at baseline can be partially explained by the high percentage of patients with delayed or unsatisfactory allograft function (55% of 45 with early TMA).

## Discussion

Our study emphasizes the poor renal allograft outcomes of transplant recipients with TMA. Moreover, the usual presence of multiple overlapping triggers supports the hypothesis that, in post transplant setting, several factors can act synergistically to injure the graft endothelium. We also highlight that ABMR is the most important limiting factor for graft survival in patients with TMA.

The low incidence of post-transplant TMA in our center is in accordance with what was reported in larger series as the study of Reynolds *et al* [4]. Differences in TMA diagnostic criteria- based on clinical, laboratorial or histological grounds- and patients' selection- percentage of sensitized recipients and with HUS as cause of primary disease- could explain the wide range of incidence rates seen in the literature. [2–6]

**Table 2. Clinical features of TMA onset (n = 89).**

| Variables | Values |
|---|---|
| Post transplant months, median (range) | 3 (0–129) |
| Early onset TMA, n (%) | 45 (51)% |
| Immunosuppressive at diagnosis, n (%) | |
| *Tacrolimus/Cyclosporine + Prednisone + AZA/MPS* | 64 (72%) |
| *Tacrolimus/Cyclosporine + Prednisone + Everolimus/Sirolimus* | 12 (14%) |
| *Tacrolimus + Prednisone* | 10 (11%) |
| *Others* | 3 (3%) |
| eGFR (ml/min/1.73m$^2$), median (range) | 17 (4–67) |
| @Proteinuria $\geq$ 1g/dL/day, n (%) | 40 (56%) |
| Hemolysis, n (%) | 22 (25%) |
| *Hemoglobin levels (g/dL)—mean ± SD* | 8.7 ± 1.5 |
| *Platelet count nadir (x 1,000/mL)—mean ± SD* | 95 ± 46 |
| *Lactate dehydrogenase (UI/L)- mean ± SD* | 682 ± 460 |
| *Lactate dehydrogenase peak > 1,000UI/L, n (%)* | 6 (27%) |
| %*Schistocytes, n (%)* | 20 (95%) |
| &*Reduced haptoglobin, n (%)* | 9 (53%) |
| Concomitant acute rejection, n (%) | 30 (34%) |
| *Cellular rejection* | 18 (21%) |
| *Antibody-mediated rejection* | 9 (10%) |
| *Mixed rejection* | 3 (3%) |
| Pregnancy at diagnosis, n (%) | 3 (3%) |
| Concomitant infection, n (%) | 48 (54%) |
| *Urinary tract infection* | 11 (13%) |
| *Cytomegalovirus* | 5 (6%) |
| *Bloodstream infection* | 3 (3%) |
| *Pneumonia* | 4 (5%) |
| *Donor-derived infection* | 3 (3%) |
| *Gastrointestinal* | 2 (2%) |
| #*Others* | 9 (10%) |
| *More than one site* | 11 (12%) |

@ out of 72 patients whose proteinuria data was available.

* out of 22 patients with hemolysis

% out of 21 patients tested for schistocyte

& out of 17 patients tested for haptoglobin

# syphilis (n = 1), tuberculosis (n = 1), cryptococcosis (n = 2), dental abscess (n = 1), polyoma virus (n = 3), empirical infection treatment (n = 4).

TMA = thrombotic microangiopathy; AZA = azathioprine; MPS = mycophenolate sodium; eGFR = estimated Glomerular Filtrate Rate; SD = standard deviation.

The baseline characteristics of our cohort, composed mainly by young and female recipients that received a kidney from a deceased donor, corroborate the results previously published [4,5,25] that suggest these are risk factors for development of post-transplant TMA. Genetic and hormonal factors could explain the particular susceptibility of these individuals to TMA in an environment with other endothelial aggressors as CNI and ischemia-reperfusion injury. [9] We caution that, due the high proportion of patients with primary kidney disease of unknown etiology in our data, we cannot fully ascertain whether some cases of recurrent aHUS were misdiagnosed as *de novo* TMA.

**Table 3. Histopathological features of TMA cases (N = 89).**

| Variable | Values |
|---|---|
| Number of glomerulus, mean ± SD | 12 ± 6 |
| Mesangiolysis, n (%) | 28 (31%) |
| Thrombi location, n (%) | |
| *Glomerular* | 44 (50%) |
| *Arteriolar* | 26 (29%) |
| *Both* | 19 (21%) |
| Morphological presentation, n (%) | |
| *Endothelial cell activation* | 54 (61%) |
| *Only thrombotic* | 35 (39%) |
| Acute Rejection, n (%) | |
| *None* | 59 (66%) |
| *Cellular rejection 1A* | 3 (3%) |
| *1B* | 7 (8%) |
| *2A* | 4 (5%) |
| *2B* | 4 (5%) |
| *Antibody-mediated rejection* | 9 (10%) |
| *Mixed rejection* | 3 (3%) |
| C4d, n (%) | |
| *Immunofluorescence positive* | 13 (15%) |
| *Immunohistochemistry negative* | 22 (25%) |
| *Immunofluorescence negative* | 33 (37%) |
| *Not performed* | 21 (23%) |
| Tubular atrophy and interstitial fibrosis, n (%) | |
| *None* | 3 (3%) |
| *Mild* | 45 (51%) |
| *Moderate* | 36 (41%) |
| *Severe* | 5 (6%) |

TMA = thrombotic microangiopathy; SD = standard deviation.

In a slim majority of the patients, TMA occurred in the early period, although there was a great variability of time of onset. These findings are in general agreement with the observations of Reynolds et al [4] that shows that, in spite of the fact that the incidence peak of *de novo* and recurrent TMA occurs in the first 6 months, the risk continues afterward. Theses results should be interpreted as an alert that TMA can be cause of allograft dysfunction at any period of post-transplantation. Furthermore, in our cohort, the timing of TMA diagnosis was not correlated to difference in rate of graft loss, in contrast to what published elsewhere [25].

Systemic TMA occurred in less than one third of the case, which is consistent with study selection criteria based in biopsy-proven TMA. The presence of hemolysis was not associated with graft failure or worse graft function, likewise the data published by Schwimmer et al. [3] It is unclear whether this is because systemic TMA is associated with more severe dysfunction at the moment of the diagnosis, leading to an earlier biopsy, diagnosis and therapy.

The evidence of ABMR associated to TMA was less common at our institution, compared to what described in other series (Satoskar et al, 55%, Wu et al 52% vs. 13%). [2,26] This can be explained by the low percentage of sensitized and re-transplanted in our patients. Nevertheless, patients who had ABMR were significantly more likely to have graft failure, as highlighted before by Wu et al. [26]

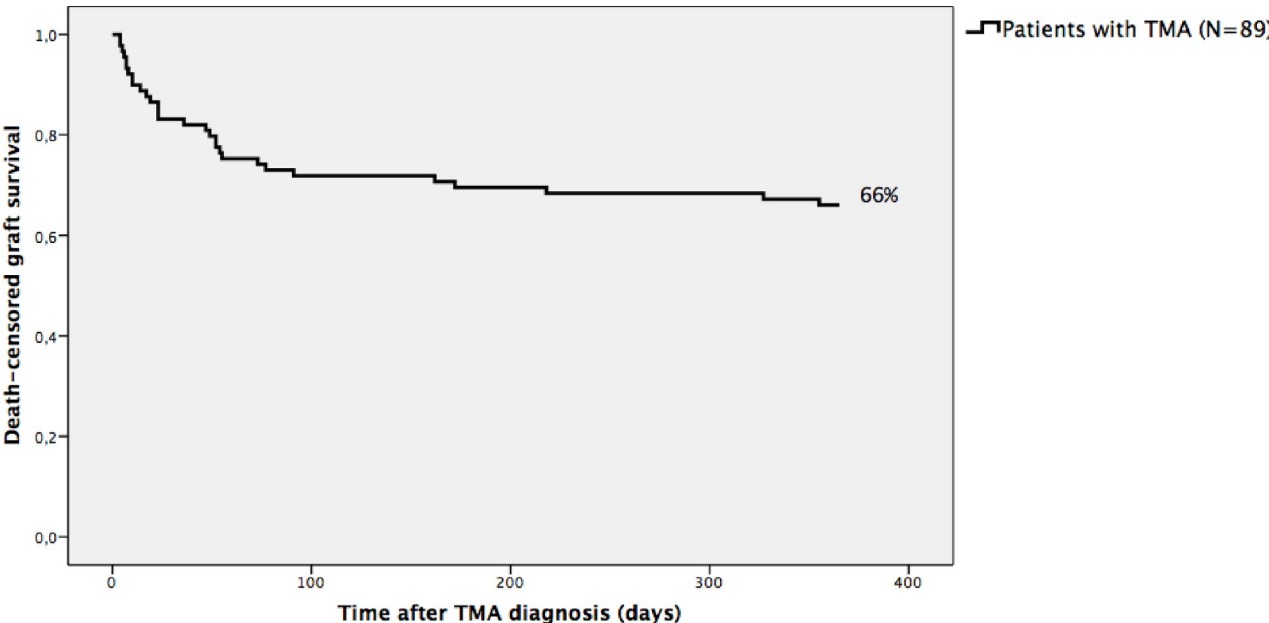

**Fig 2. Kaplan-Meier death-censored graft survival of kidney or pancreas-kidney transplanted patients with TMA (n = 89).** Survival outcomes were analysed after one-year of TMA diagnosis. TMA = Thrombotic microangiopathy.

Beyond that, despite the prevalence of pre-transplant AT1R antibodies in serum samples of our patients was similar to what was published in literature, two patients had an ABMR-TMA due to AT1R antibodies emphasizing the hypothesis that these non-HLA (human leukocyte

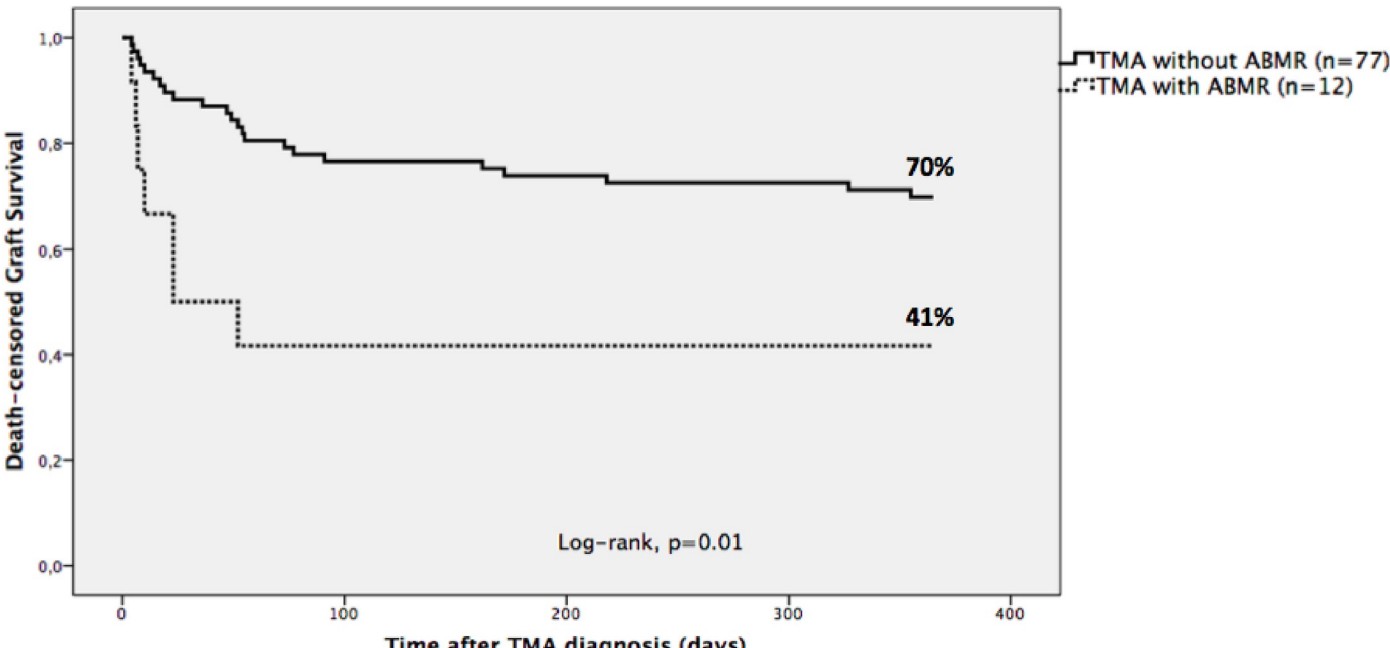

**Fig 3. Kaplan-Meier death-censored graft survival of kidney or pancreas-kidney transplanted patients with TMA associated or not with ABMR.** Patients with TMA associated with ABMR had a statistically significant higher incidence of graft loss (with ABMR vs. without ABMR, p = 0.01-log-rank test). TMA = Thrombotic microangiopathy; ABMR = antibody-mediated rejection.

**Table 4. Evolution of eGFR in time of patients with TMA (mean±SD).**

| | Timeline | | | |
|---|---|---|---|---|
| | **Baseline** | **TMA Diagnosis** | **One-year after** | **p** |
| Total (n = 89) | 36,94 ± 25,94[a'] | 20,64 ± 15,46[b'] | 28,64 ± 23,69[b'] | <0,001§ |
| Hemolysis | | | | |
| with (n = 22) | 39,16 ± 25,53[a'] | 22,15 ± 15,10[b'] | 29,11 ± 23,82[b'] | <0,001§ |
| without (n = 67) | 30,20 ± 26,58[a'] | 16,06 ± 16,01[b'] | 27,20 ± 23,77 | 0,007§ |
| Onset | | | | <0,001 |
| Early (n = 45) | 27,08 ± 25,40[a] | 17,16 ± 15,42[b] | 30,92 ± 27,01[a] | |
| Late (n = 44) | 47,02 ± 22,61[a] | 24,20 ± 14,85[b] | 26,32 ± 19,77[b] | |
| Antibody-mediated rejection | | | | |
| with (n = 12) | 38,69 ± 25,76[a'] | 21,88 ± 15,58[b'] | 29,19 ± 22,38[b'] | <0,001§ |
| without (n = 77) | 25,72 ± 25,28[a'] | 12,71 ± 12,52[b'] | 25,13 ± 31,85[a'] | 0,011§ |
| Thrombi location | | | | <0,001 |
| Glomerular (n = 44) | 37,47 ± 22,49[a] | 24,24 ± 14,53[c] | 27,47 ± 20,41[b] | |
| Arteriolar (n = 26) | 41,83 ± 28,20[a] | 20,92 ± 15,41[c] | 33,32 ± 25,38[b] | |
| Both (n = 19) | 24,89 ± 21,78[a] | 15,08 ± 16,02[c] | 19,41 ± 21,91[b] | |
| Endothelial cell activation | | | | <0,001 |
| with (n = 54) | 44,59 ± 28,74[a] | 23,25 ± 15,64[c] | 35,21 ± 25,64[b] | |
| without (n = 35) | 31,99 ± 22,88[a] | 18,95 ± 15,25[c] | 24,39 ± 21,52[b] | |

p—Time effect by ANOVA or Friedman nonparametric test (§)

a, b e c—ANOVA (post-hoc test Bonferroni) differences between mean values in time.

a', b' e c'—Friedman / Mann-Whitney or Kruskal-Wallis (post-hoc test Dunn-Bonferroni) differences between mean values in time.

TMA = thrombotic microangiopathy; eGFR = estimated Glomerular Filtrate Rate; SD = standard deviation

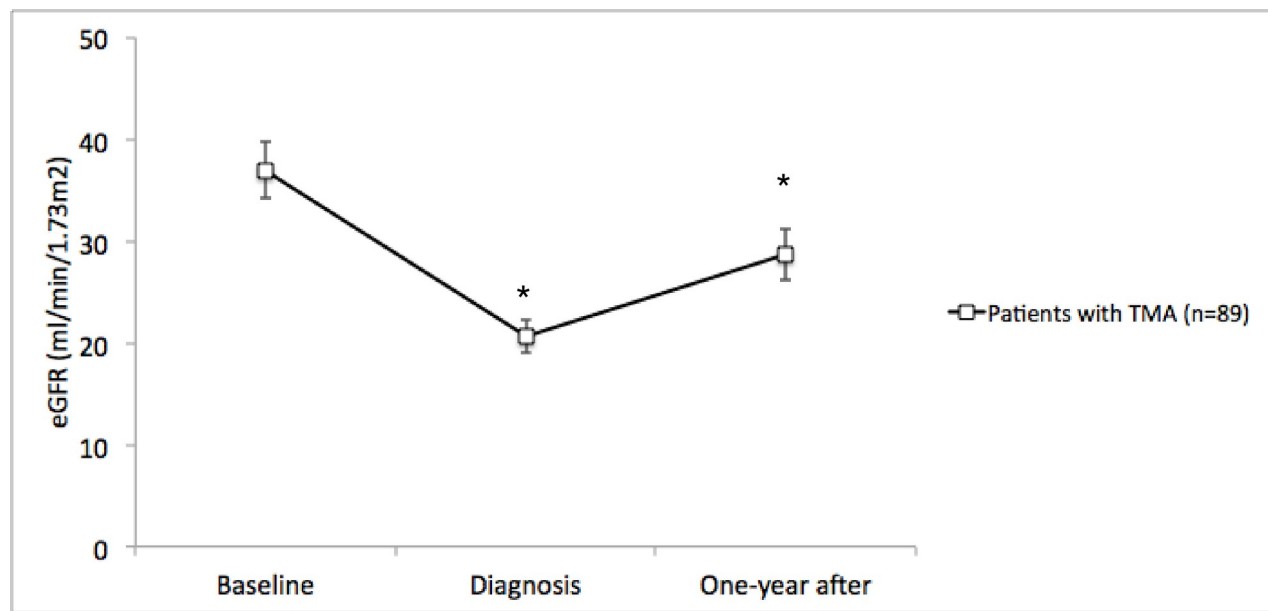

**Fig 4. Allograft function of kidney-transplanted patients with TMA at baseline, TMA diagnosis and after one-year of follow-up (n = 89).** A statistically significant drop of mean eGFR±SE occurred at diagnosis (Baseline vs. Diagnosis, p<0.001), which was permanent until the end of the study (Diagnosis vs. One-year after, p = 0.13; Baseline vs. One-year after, p<0.001). Baseline eGFR was calculated with the lowest serum creatinine level in up to 3 months before TMA diagnosis and patients with graft failure were considered to have an eGFR of 5 ml/min/1,73m$^2$. * means p< 0.05 in relation to baseline eGFR. TMA = Thrombotic microangiopathy; eGFR = estimated glomerular filtration rate; SE = standard error.

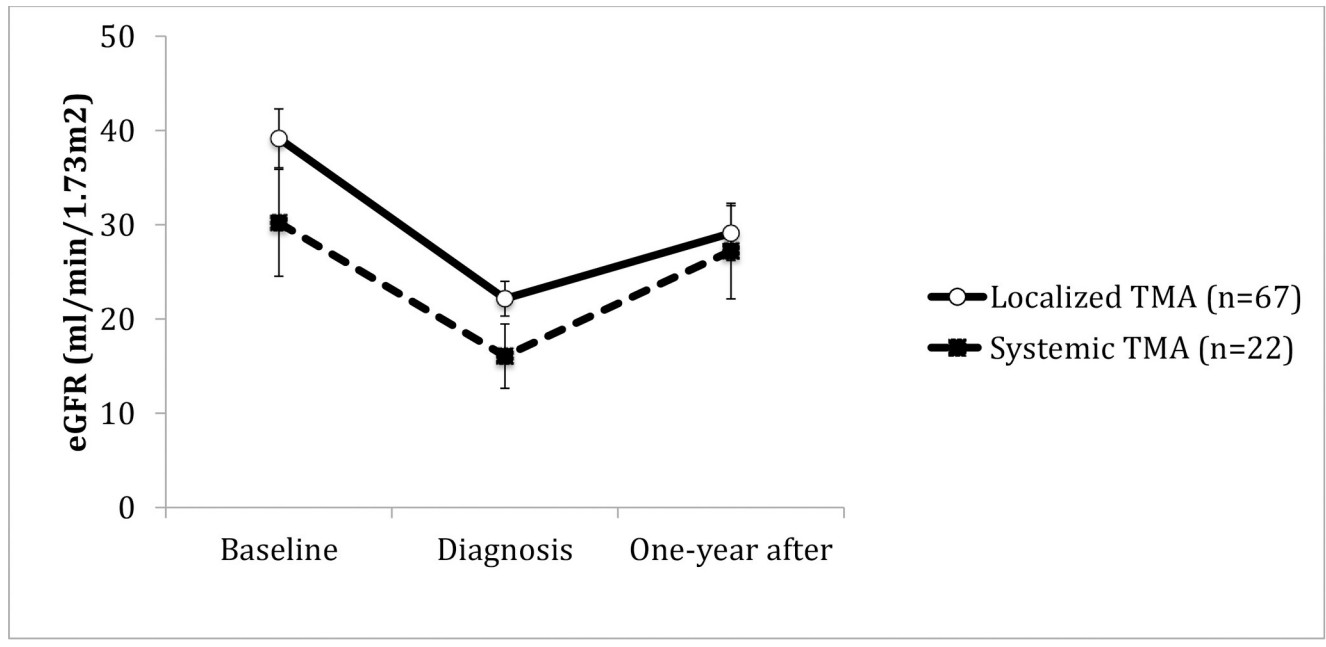

**Fig 5. Allograft function of kidney-transplanted patients with TMA at baseline, TMA diagnosis and after one-year of follow-up, according to the presence of hemolysis.** Hemolysis was associated with a lower mean eGFR±SE at TMA diagnosis (Systemic vs. Localized TMA at diagnosis- p = 0.04), however at the end of follow-up, there was no difference between groups (Systemic vs. Localized TMA one-year after- p = 0.76). Systemic TMA was defined by the presence of anemia, plaquetopenia and DHL> 1,000 U/L or schistocyte or reduced haptoglobin. Baseline eGFR was calculated with the lowest serum creatinine level in up to 3 months before TMA diagnosis and patients with graft failure were considered to have an eGFR of 5 ml/min/1,73m$^2$. TMA = Thrombotic microangiopathy; eGFR = estimated glomerular filtration rate; SE = standard error.

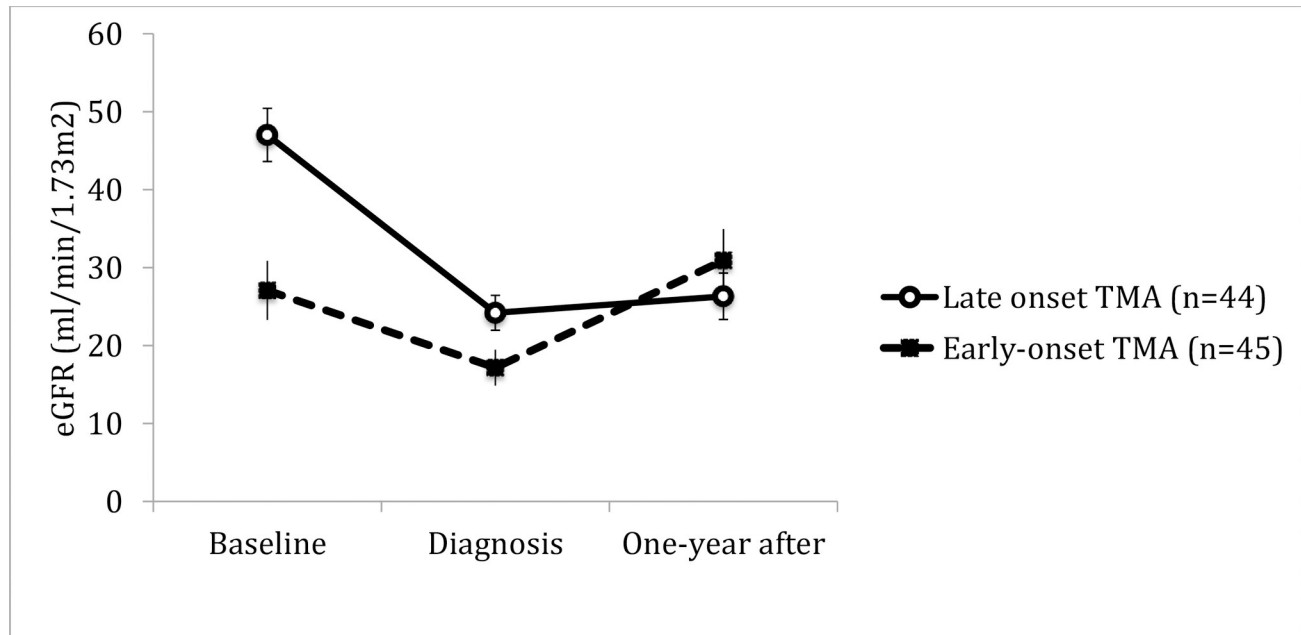

**Fig 6. Allograft function of kidney-transplanted patients with TMA at baseline, TMA diagnosis and after one-year of follow-up, according to the time of onset.** Patients with a early onset TMA had a lower eGFR±SE at baseline (Early onset vs. Late onset TMA at baseline, p = <0.001), nevertheless, after one year following TMA diagnosis the allograft function was similar in both groups (Early onset vs. Late onset TMA one-year after, p = 0.31) Early onset TMA was defined when it occurred less than 3 months post-transplant. Baseline eGFR was calculated with the lowest serum creatinine level in up to 3 months before TMA diagnosis and patients with graft failure were considered to have an eGFR of 5 ml/min/1,73m$^2$. TMA = Thrombotic microangiopathy; eGFR = estimated glomerular filtration rate; SE = standard error.

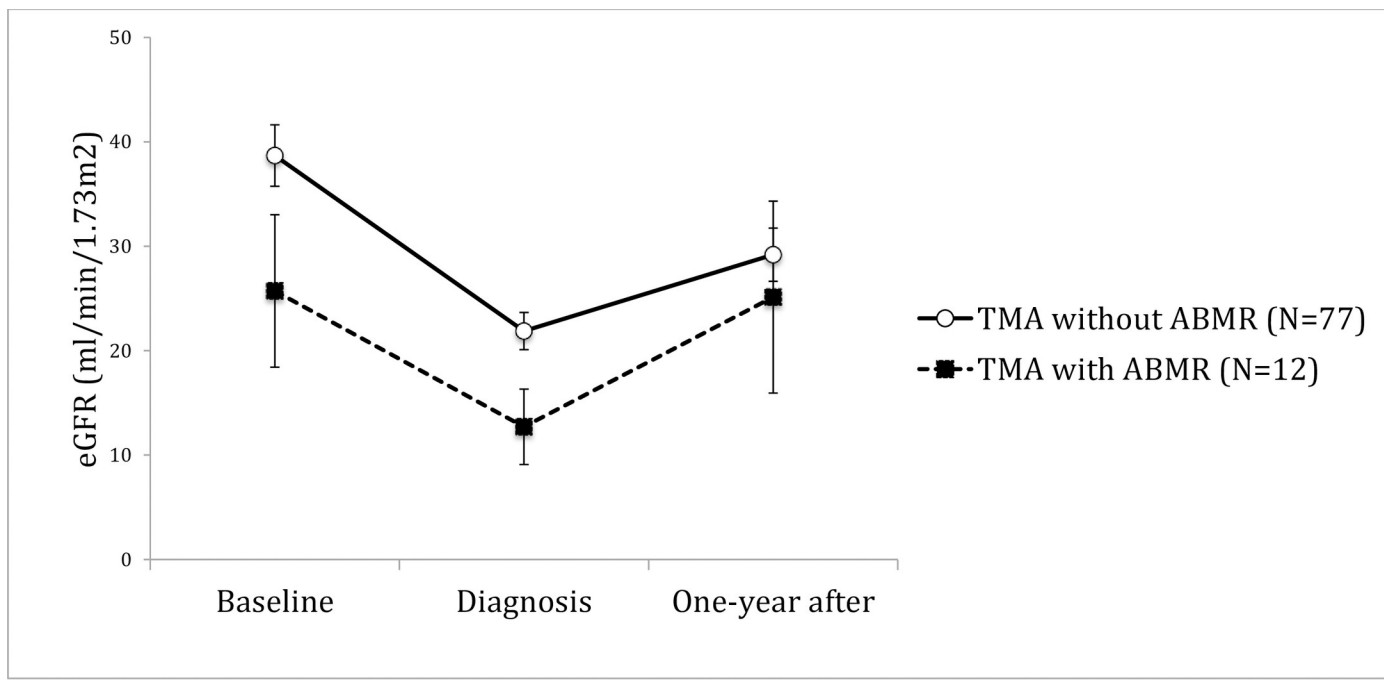

**Fig 7. Allograft function of kidney-transplanted patients with TMA at baseline, TMA diagnosis and after one-year of follow-up, according to the presence of ABMR.** Patients with TMA associated to ABMR had a mean eGFR lower at baseline, which was persistent until the end of follow-up (p = 0.08). On follow-up, the allograft function was similar in patients with or without AMR (p = 0.257). Baseline eGFR was calculated with the lowest serum creatinine level in up to 3 months before TMA diagnosis and patients with graft failure were considered to have an eGFR of 5 ml/min/1,73m$^2$. TMA = Thrombotic microangiopathy; ABMR = Antibody Mediated Rejection; eGFR = estimated glomerular filtration rate; SE = standard error.

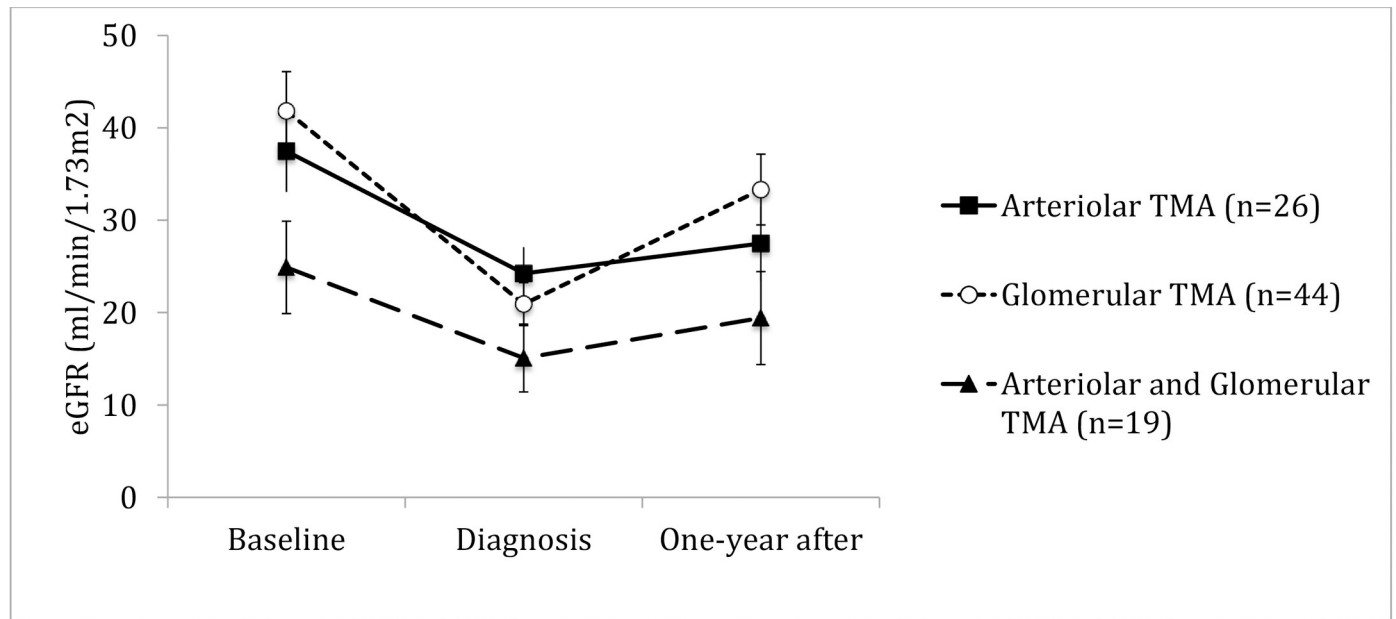

**Fig 8. Allograft function of kidney-transplanted patients with TMA at baseline, TMA diagnosis and after one-year of follow-up, according to the thrombi location.** Patients with thrombi located in both arterioles and glomerulus had a lower eGFR±SE than those whose thrombi located only in arterioles at all moments of observation (Arteriolar TMA vs. Arteriolar and Glomerular TMA, p = 0.008). TMA glomerular was defined when thrombi was located in afferent and efferent arteriole or glomerular capillary and TMA arteriolar, when it was located in arterioles or interlobular arteries. Baseline eGFR was calculated with the lowest serum creatinine level in up to 3 months before TMA diagnosis and patients with graft failure were considered to have an eGFR of 5 ml/min/1,73m$^2$. TMA = Thrombotic microangiopathy; eGFR = estimated glomerular filtration rate; SE = standard error.

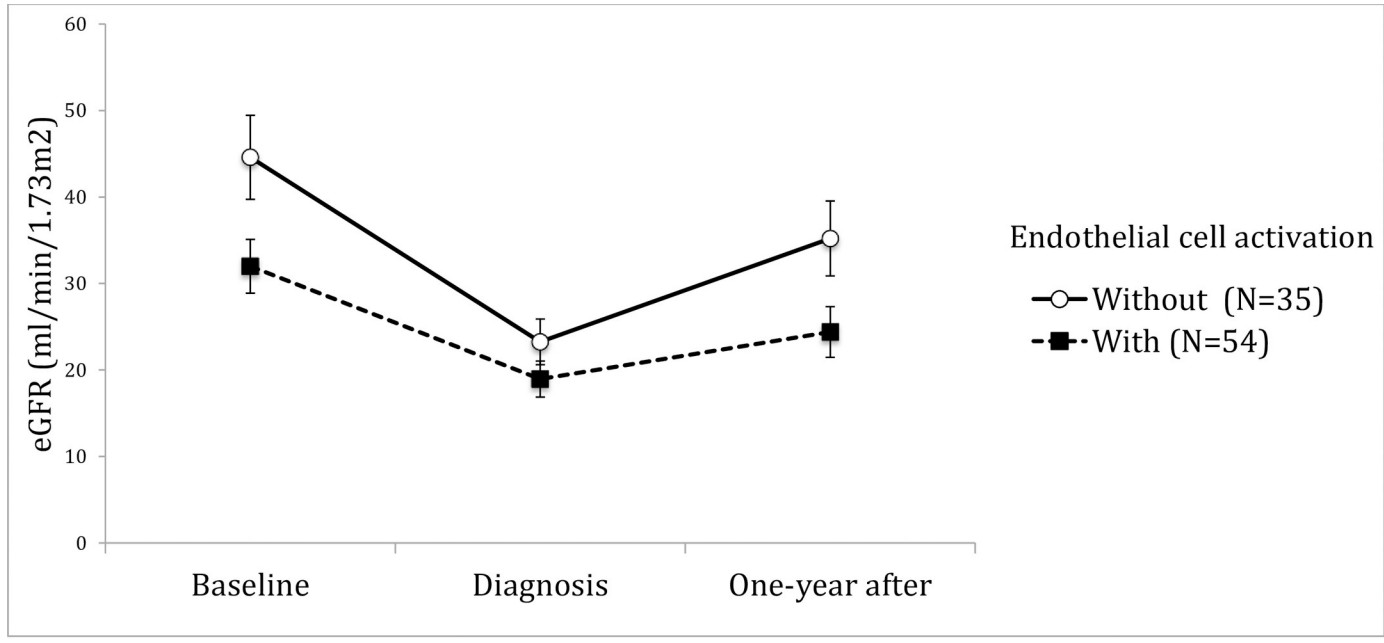

**Fig 9. Allograft function of kidney-transplanted patients with TMA at baseline, TMA diagnosis and after one-year of follow-up, according to presence endothelial cell activation (EC activation).** Patients with TMA lesions with EC activation had a lower eGFR±SE, at all moments of observation (with vs. without, p = 0.013). EC activation was defined when there was mesangiolysis, capillary necrosis, glomerular endothelial detachment, capillary wall thickening obliterative arteriolopathy defined as luminal occlusion with mural myxoid or fibrinoid change, thickening of the vessel wall. Baseline eGFR was calculated with the lowest serum creatinine level in up to 3 months before TMA diagnosis and patients with graft failure were considered to have an eGFR of 5 ml/min/1,73m². TMA = Thrombotic microangiopathy; eGFR = estimated glomerular filtration rate; SE = standard error.

antigen) antibodies may be associated to microvascular inflammation, early acute rejection and allograft loss as previously reported. [27–29]

TMA associated only to drug toxicity was relatively infrequent in our findings, compared to what was observed by Nava *et al.* [30]. On the other hand, in accordance with what was recently published by Bayer *et al* [31], the usual presence of multiples conditions causing or precipitating TMA supports the "multiple hit hypothesis" [8,32] that speculates that TMA is the consequence of the combination of genetic predisposition and several trigger factors/conditions in both native and transplanted kidneys.

Although eculizumab, a C5-targeted complement blocker, is very promise in prophylaxis and treatment for recurrent aHUS after kidney transplantation [33,34], it was not a therapeutic option in our cohort, since, in Brazil's publicly funded health care system, it is only available for treatment of patients with paroxysmal nocturnal hemoglobinuria (PNH). [35,36] The high cost of medication and need for prolonged treatment preclude the financing of this therapy by the local transplant centers or the patient itself.

The high rate of renal allograft loss in our cohort is probably related to the unspecific therapeutic approach performed, explained by the unavailability of complement blockers and diagnostic tools for differentiation of the etiologies and triggers of post-transplant TMA.

Regarding the histological features, the thrombi location had no predictive value for graft failure in our study, which is consistent with the results of Satoskar *et al* [2] but is in disagreement with what have been published by Wu et *al* [25], where the pattern and severity of vasculopathy of TMA were associated to poor allograft outcomes. The presence of endothelial cell activation in the allograft biopsy diagnostic of TMA was also not associated to a higher risk of allograft loss. We hypothesize that endothelial damage probably is a more determinant matter

if it occurs in a chronic and continuous fashion as seen in transplant glomerulopathy [6,37,38].

This study has some limitations inherent to its retrospective methodology, such as rigorous diagnosis workup not always available, selection criteria based on histological TMA diagnosis and small sample sized, that precludes drawing definitive conclusions. Other issue that needs consideration is the lack of complement mutational analysis. However it would be interesting to reveal the individual predisposing factors, current evidence does not support evaluation of the complement system in all patients with *de novo* TMA [11]. On the other hand, our study is the first to relate several clinical and histological features of biopsy-proven TMA in kidney-transplanted patients.

In summary, our data suggest that TMA is a rare but severe condition in the setting of renal transplant, regardless of its clinical or pathological presentation. When associated to ABMR, its prognosis is even worse. It is important to recognize that the lack of a tailored therapeutic strategy, as the complement blockade, partially accounts for the bad outcomes of our findings. Further prospective studies carefully looking at the impact of this treatment are required to test this hypothesis.

## Acknowledgments

The authors gratefully thank the patients and nursing staff who participated in this study.

## Author Contributions

**Conceptualization:** Hélio Tedesco Silva Junior.

**Data curation:** Cínthia Montenegro Teixeira.

**Formal analysis:** Cínthia Montenegro Teixeira.

**Funding acquisition:** Cínthia Montenegro Teixeira.

**Investigation:** Cínthia Montenegro Teixeira.

**Methodology:** Hélio Tedesco Silva Junior, Luiz Antônio Ribeiro de Moura, Henrique Machado de Sousa Proença, Renato de Marco, Maria Gerbase de Lima, Marina Pontello Cristelli, Laila Almeida Viana, Cláudia Rosso Felipe, José Osmar Medina Pestana.

**Project administration:** Hélio Tedesco Silva Junior, José Osmar Medina Pestana.

**Resources:** Luiz Antônio Ribeiro de Moura, Henrique Machado de Sousa Proença, Renato de Marco, Maria Gerbase de Lima, Marina Pontello Cristelli, Laila Almeida Viana, Cláudia Rosso Felipe, José Osmar Medina Pestana.

**Supervision:** Hélio Tedesco Silva Junior, Luiz Antônio Ribeiro de Moura, Henrique Machado de Sousa Proença, Renato de Marco, Maria Gerbase de Lima, Marina Pontello Cristelli, Laila Almeida Viana, Cláudia Rosso Felipe, José Osmar Medina Pestana.

**Validation:** Hélio Tedesco Silva Junior.

**Visualization:** Hélio Tedesco Silva Junior.

**Writing – original draft:** Cínthia Montenegro Teixeira.

**Writing – review & editing:** Cínthia Montenegro Teixeira, Luiz Antônio Ribeiro de Moura, Henrique Machado de Sousa Proença, Renato de Marco, Maria Gerbase de Lima, Marina Pontello Cristelli, Laila Almeida Viana, Cláudia Rosso Felipe, José Osmar Medina Pestana.

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
