## [Decision Letter · Decision Letter 0]

2 Oct 2019

PONE-D-19-23294

Clinical and pathological features of thrombotic microangiopathy influencing long-term kidney transplant outcomes

PLOS ONE

Dear Mrs Teixeira,

Thank you for submitting your manuscript to PLOS ONE. After careful consideration, we feel that it has merit but does not fully meet PLOS ONE’s publication criteria as it currently stands. Therefore, we invite you to submit a revised version of the manuscript that addresses the points raised during the review process.

**The manuscript focuses on a topic of potential interest. There are, however, few shortcomings in the study the authors should address. To mention few of them, i) need to expand more the conclusions; ii) need to discuss about the high rate of allograft loss and potential causes of this; iii) need to include in the method section how the cases are reviewed for the diagnosis of TMA prior to the confirmation by reviewing kidney biopsies; iv) need to include the methods of their search such as "CPT codes" used; v) need to clarify if dialysis was temporary; vi) concern about the statement that their cohort suggest risk factors for development of de novo TMA; vii) need to implement the discussion about eculizumab therapy and its use in Brazil, if any**.

We would appreciate receiving your revised manuscript by Nov 16 2019 11:59PM. To enhance the reproducibility of your results, we recommend that if applicable you deposit your laboratory protocols in protocols.io, where a protocol can be assigned its own identifier (DOI) such that it can be cited independently in the future. For instructions see: http://journals.plos.org/plosone/s/submission-guidelines#loc-laboratory-protocols

We look forward to receiving your revised manuscript.

Kind regards,

Giuseppe Remuzzi

Academic Editor

PLOS ONE

Journal Requirements:

1. In ethics statement in the manuscript and in the online submission form, please provide additional information about the patient records used in your retrospective study. Specifically, please ensure that you have discussed whether all data were fully anonymized before you accessed them and/or whether the IRB or ethics committee waived the requirement for informed consent. If patients provided informed written consent to have data from their medical records used in research, please include this information.

2. Thank you for including your ethics statement: The study protocol was approved by the institutional review board (CEP-UNIFESP-parecer 1643995) and adheres to the 2000 Declaration of Helsinki as well as the Declaration of Istanbul 2008. The consent was not obtained because the data were analyzed retrospectively and anonymously.

3. Thank you for including your competing interests statement; "I have read the journal's policy and the authors of this manuscript have the following competing interests: Dr. HTS reports grants and personal fees from NOVARTIS, grants and personal fees from PFIZER, grants and personal fees from SANOFI and grants from QUARK, outside the submitted work. All the other authors declared no relevant competing interests. "

4. Please include a copy of Table 5 which you refer to in your text on page 18

Reviewers' comments:

Reviewer's Responses to Questions

**Comments to the Author**

1. Is the manuscript technically sound, and do the data support the conclusions?

Reviewer #1: Yes

Reviewer #2: Yes

2. Has the statistical analysis been performed appropriately and rigorously? 

Reviewer #1: I Don't Know

Reviewer #2: Yes

3. Have the authors made all data underlying the findings in their manuscript fully available?

Reviewer #1: Yes

Reviewer #2: Yes

4. Is the manuscript presented in an intelligible fashion and written in standard English?

Reviewer #1: Yes

Reviewer #2: Yes

5. Review Comments to the Author

Reviewer #1: Line 71 and Line 510: I would change the verb "prove" to "suggest" in the conclusion. And it would be beneficial for the authors to expand more on their conclusions. They should discuss about the high rate of allograft loss and potential causes of this.

In the methods section, they should include how the cases were reviewed for the diagnosis of TMA prior to the confirmation by reviewing kidney biopsies, and include the methods of their search such as "CPT codes" used or explained how they found/ collected 119 cases with diagnosis of TMA (how was this original diagnosis made).

And this should be included as a limitation in the discussion part.

Line 248, Authors mention that over one third of patients were on dialysis, please clarify if dialysis was temporary or if there was allograft loss within the first 3 months post transplant.

Lines 446-449. Authors imply that their cohort suggest risk factors for development of de novo TMA, but without not know the primary disease of ESRD 37% on the cohort, it is hard to make a conclusion of de novo presentation. This should be clarified in this paragraph.

In the discussion, the authors should also include the lack of therapy with eculizumab in their cohort, and should recognize this drug as part of the therapy for aHUS worldwide, and perhaps disclose if this medication is not routinely used in Brazil at the moment (due to cost or other reasons, this should also be explained)

Reviewer #2: The authors have conducted a thorough retrospective review of 89 kidney transplant patients with TMA and classified their clinical and pathological findings in detail. the main observation of the study is the TMA associated with ABMR has a considerably poorer prognosis than TMA not associated with ABMR. The pathologic features did not play as large a role in differentiating outcomes with respect to dGFR.

6. PLOS authors have the option to publish the peer review history of their article (what does this mean?). If published, this will include your full peer review and any attached files.

Reviewer #1: No

Reviewer #2: Yes: Stuart J. Knechtle

---

## [Author Response · Author response to Decision Letter 0]

24 Nov 2019

Dear Academic Editor and reviewers,

Please find enclosed the corrected version of the manuscript entitled “Clinical and pathological features of thrombotic microangiopathy influencing long-term kidney transplant outcomes”. The answers to the reviewers are disposed bellow.

1. need to expand more the conclusions

It is important to recognize that the lack of a tailored therapeutic strategy, as the complement blockade, partially accounts for the bad outcomes of our findings. Further prospective studies carefully looking at the impact of this treatment are required to test this hypothesis. 

This statement was added on “Discussion” section.

2. need to discuss about the high rate of allograft loss and potential causes of this

The high rate of renal allograft loss in our cohort is probably related to the unspecific therapeutic approach performed, explained by the unavailability of complement blockers and diagnostic tools for differentiation of the etiologies and triggers of post-transplant TMA. 

This statement was added on “Discussion” section.

3. need to clarify if dialysis was temporary

At the time of TMA diagnosis, 36% of the patients (n=32) were on dialysis. There was allograft loss within the first 3 months post transplant in 13 patients. 

This information was written in “Results” section.

4. concern about the statement that their cohort suggest risk factors for development of de novo TMA

 We caution that, due the high proportion of patients with primary kidney disease of unknown etiology in our data, we cannot fully ascertain whether some cases of recurrent aHUS were misdiagnosed as de novo TMA. 

We added this caveat on “Discussion” section and suppressed the expression “de novo” of the sentence on line 457.

5. need to implement the discussion about eculizumab therapy and its use in Brazil, if any.

Although eculizumab was not a therapeutic option in our cohort, since, in Brazil's publicly funded health care system, it is only available for treatment of patients with paroxysmal nocturnal hemoglobinuria (PNH). The high cost of medication and need for prolonged treatment preclude the financing of this therapy by the local transplant centers or the patient itself. 

We addressed this issue on “Discussion” section.

6. Review Line 71 and Line 510: I would change the verb "prove" to "suggest" in the conclusion.

Thank you. We changed the verb.

7. In the methods section, they should include how the cases were reviewed for the diagnosis of TMA prior to the confirmation by reviewing kidney biopsies, and include the methods of their search such as "CPT codes" used or explained how they found/ collected 119 cases with diagnosis of TMA (how was this original diagnosis made).

And this should be included as a limitation in the discussion part.

In Brazil or Hospital do Rim, the search for CPT codes is not available. In this retrospective cohort study, we initially retrieved all consecutive unselected reports of renal transplant biopsies from Hospital do Rim database between January 2011 and December 2015. These biopsies were performed for graft dysfunction, new onset proteinuria or delayed graft function from kidney and kidney-pancreas transplanted patients. Of a total of 6886, we selected 119 biopsies whose reports described features of TMA. Final diagnosis was confirmed by one of the pathologist authors (LARM). 

We added this information on “Methodology” section. In addition, we wrote a statement on “Discussion” section addressing the possible selection criteria bias as a limitation of our study, since TMA diagnosis was based only on histological criteria.

8. Please include a copy of Table 5 which you refer to in your text on page 18

On page 18 line 367, there was a typing error: there is no Table 5. The results are related to Table 4, instead. We are sorry about it.

9. Additional requirements

We amended current and reviewed statements about ethics and competing interests in the manuscript and in the online submission form.

I hope this new version will fit the requirements for publication in PLOS ONE.

I am looking forward to hearing from you soon.

Sincerely,

Dr. Cínthia Montenegro Teixeira, the corresponding author

---

## [Decision Letter · Decision Letter 1]

19 Dec 2019

Clinical and pathological features of thrombotic microangiopathy influencing long-term kidney transplant outcomes

PONE-D-19-23294R1

Dear Dr. Teixeira,

We are pleased to inform you that your manuscript has been judged scientifically suitable for publication and will be formally accepted for publication once it complies with all outstanding technical requirements.

The revised version of the manuscript is definitely improved. The authors have adequately addressed all the reviewers’ comments.

With kind regards,

Giuseppe Remuzzi

Academic Editor

PLOS ONE

Additional Editor Comments (optional):

Reviewers' comments:

Reviewer's Responses to Questions

**Comments to the Author**

1. If the authors have adequately addressed your comments raised in a previous round of review and you feel that this manuscript is now acceptable for publication, you may indicate that here to bypass the “Comments to the Author” section, enter your conflict of interest statement in the “Confidential to Editor” section, and submit your "Accept" recommendation.

Reviewer #1: All comments have been addressed

Reviewer #2: All comments have been addressed

2. Is the manuscript technically sound, and do the data support the conclusions?

Reviewer #1: Yes

Reviewer #2: Yes

3. Has the statistical analysis been performed appropriately and rigorously? 

Reviewer #1: Yes

Reviewer #2: Yes

4. Have the authors made all data underlying the findings in their manuscript fully available?

Reviewer #1: Yes

Reviewer #2: Yes

5. Is the manuscript presented in an intelligible fashion and written in standard English?

Reviewer #1: Yes

Reviewer #2: Yes

6. Review Comments to the Author

Reviewer #1: Very interesting topic. Authors have answered in detail comments made by this reviewer. I have no other comments to add at this moment.

Reviewer #2: There are several examples of words that are used incorrectly and need to be edited, but the content of the paper is excellent and the authors have responded to the reviewers' concerns.

7. PLOS authors have the option to publish the peer review history of their article (what does this mean?). If published, this will include your full peer review and any attached files.

Reviewer #1: No

Reviewer #2: Yes: Stuart Knechtle

---

## [Editor Report · Acceptance letter]

27 Dec 2019

PONE-D-19-23294R1 

Clinical and pathological features of thrombotic microangiopathy influencing long-term kidney transplant outcomes 

Dear Dr. Teixeira:

I am pleased to inform you that your manuscript has been deemed suitable for publication in PLOS ONE. Congratulations! Your manuscript is now with our production department. 

With kind regards,

on behalf of

Prof. Giuseppe Remuzzi 

Academic Editor

PLOS ONE